# Ultrasound Evaluation of the Primary α Phase Grain Size Based on Generative Adversarial Network

**DOI:** 10.3390/s22093274

**Published:** 2022-04-24

**Authors:** Siqin Peng, Xi Chen, Guanhua Wu, Ming Li, Hao Chen

**Affiliations:** 1Key Laboratory of Nondestructive Test of Ministry of Education, Nanchang Hangkong University, Nanchang 330063, China; psq0079@outlook.com (S.P.); chenxi@nchu.edu.cn (X.C.); wuguanhua@nchu.edu.cn (G.W.); liming@nchu.edu.cn (M.L.); 2School of Information Engineering, Nanchang Hangkong University, Nanchang 330063, China

**Keywords:** virtual samples, GAN, titanium alloy, primary α phase, ultrasound evaluation

## Abstract

Because of the high cost of experimental data acquisition, the limited size of the sample set available when conducting tissue structure ultrasound evaluation can cause the evaluation model to have low accuracy. To address such a small-sample problem, the sample set size can be expanded by using virtual samples. In this study, an ultrasound evaluation method for the primary α phase grain size based on the generation of virtual samples by a generative adversarial network (GAN) was developed. TC25 titanium alloy forgings were treated as the research object. Virtual samples were generated by the GAN with a fully connected network of different sizes used as the generator and discriminator. A virtual sample screening mechanism was constructed to obtain the virtual sample set, taking the optimization rate as the validity criterion. Moreover, an ultrasound evaluation optimization problem was constructed with accuracy as the target. It was solved by using support vector machine regression to obtain the final ultrasound evaluation model. A benchmark function was adopted to verify the effectiveness of the method, and a series of experiments and comparison experiments were performed on the ultrasound evaluation model using test samples. The results show that the learning accuracy of the original small samples can be increased by effective virtual samples. The ultrasound evaluation model built based on the proposed method has a higher accuracy and better stability than other models.

## 1. Introduction

For large titanium alloy forgings, because of their high cost, the metallographic analysis method with accompanying test specimens and sample dissection, as well as full-coverage ultrasonic testing and noise wave height threshold evaluation, is usually used for quality inspection and testing [1,2,3]. Titanium alloys have a complex structure, and the noise wave height threshold method of ultrasonic inspection is based on a single judgment. Hence, it is difficult to meet the quality inspection requirements resulting from the upgrading of the aero-engine manufacturing process in China. To address this issue, quantitative methods for ultrasound evaluation of the titanium alloy metallographic structure are needed. TC25 titanium alloy is an α+β type heat-strength titanium alloy with good high-temperature strength, thermal stability, and corrosion resistance, and it is an ideal material for manufacturing aero-engines [4]. The most important microstructure parameters for the mechanical properties of dual-mode titanium alloys are the size of the primary α phase and its volume content, and the volume of the primary α phase is closely related to the size. Therefore, the primary α phase is the key metallographic organization of the dual-phase titanium alloy, and its quantitative detection can be used to evaluate the manufacturing quality of effectively [5].

The scattering of ultrasonic waves by the titanium alloy tissue structure is the foundation for performing the ultrasound evaluation. Blodgett [6] and Yang et al. [7,8] studied the scattering mechanism of ultrasonic waves by duplex titanium alloy tissue structure, and they analyzed the effects of the titanium alloy tissue structure and texture on such acoustic characteristics as ultrasonic attenuation, backscattering, and the nonlinear coefficient. They also established various models of the relationship between titanium alloy structure characterization model parameters and ultrasonic attenuation and backscattering. These relationship models were verified experimentally.

Limited by the adjustable range and gradient of thermal processing parameters, test specimen processing cost, and other factors, the number of experimental samples is restricted in the ultrasound evaluation of metal structures. Yang et al. used one to four samples for validation [7,8]. Li used six to nine samples for model construction and verification in a study of the ultrasound evaluation of metal grain size [9,10]. Both the size of the sample set and the amount of information contained in the sample affect the effectiveness of the evaluation model [11,12]. Therefore, improving the small sample space is of great value to optimize the performance of prediction models [13,14,15].

Generating virtual samples is one of the main means to solve the problem of small samples. Poggio et al. [16] first proposed the concept of a virtual sample, providing ideas for solving the problem of small samples. Huang et al. [17] used the normal diffusion function based on fuzzy theory to fill the discrete information interval. On this basis, Li [18] proposed mega-trend-diffusion (MTD). Chen et al. [19] improved the MTD method and proposed multi-distribution mega trend diffusion (MD-MTD) to increase the quality of virtual samples by dividing the sample distribution area.

The ultrasound evaluation of the grain size of the primary α phase is a typical small-sample problem. It involves the problem of low evaluation accuracy caused by the small scale of the sample set or insufficient sample information. In this study, to address the above problems, the appropriate virtual sample generation technology was selected according to the ultrasound evaluation characteristics of the grain size of primary α phase. Based on the real sample data information, an effective virtual sample screening mechanism was established to screen effective virtual samples. Furthermore, the effective virtual samples were used to expand the real sample set, and the support vector machine (SVM) regression method was adopted to establish a multiparameter ultrasonic prediction model of primary α phase grain size. The evaluation results were verified by a series of single ultrasonic feature fitting models without virtual samples and multiparameter ultrasonic prediction models using other algorithms.

## 2. Virtual Sample Generation Technology

### 2.1. Definition of Virtual Samples

Let the original training sample set be (x,y), where $x\in R^{n}$. A generation relationship *K*, produced by a priori knowledge, is used to generate some $x_{vir}$. Then, a transformation relationship $\hat{H}$ is defined to make $y_{vir}=\hat{H}(x_{vir})$, with the newly generated samples $\left(x_{vir},y_{vir}\right)$ being called “virtual samples”, where $x_{vir}$ and $y_{vir}$ are the input value and output value of virtual samples, respectively.

Virtual samples are divided into effective and invalid virtual samples. Only effective virtual samples can improve the small sample space [20] to increase the learning accuracy of the small sample set and reduce the prediction error of the model [21,22].

When building the virtual sample space, the prediction model $\hat{H}$, the prediction model of the actual small sample set, can be constructed through common machine-learning methods, such as multiple linear regression and SVM. Nevertheless, its establishment condition is that $\hat{H}$ can be used to generate y, which corresponds to the input attribute x only when the mean absolute percentage error (MAPE) of the prediction model is less than 10% [23].

### 2.2. Introduction of Virtual Samples

Poggio et al. [16] first introduced the concept of virtual samples and provided relevant ideas for solving the small sample problem. Niyogi et al. [24] proved mathematically that virtual samples constructed from a priori knowledge can provide valid information as well. Subsequently, virtual sample generation technology has been applied widely in many fields, such as image recognition, soft measurement, medical diagnosis, and fault diagnosis [25,26,27,28,29]. Virtual sample generation (VSG) methods are generally classified into three categories.

Sampling-based VSG is a process of generating random variables. This process generally involves generating new examples from the specified distribution, such as using parameter distributions, including the Weibull distribution and Gaussian distribution, to improve the data distribution of multimodal small-sample datasets [30,31]. This method is often used to synthesize new samples in unbalanced datasets [32].

Information diffusion-based VSG is a method based on diffusion theory. This method is generally used to estimate the acceptable range of small-sample-set attributes, generate new samples through the diffusion function, and fill the sample information interval, so as to expand the sample set, such as mega-trend-diffusion [18] (MTD) and multi-distribution mega-trend-diffusion [19] (MD-MTD).

Deep-learning-based VSG is a method using a deep generative model. The core idea of this method is to simulate the data distribution of real samples to generate new false sample data. The common deep generative model is the generative adversarial network [33] (GAN).

The first method has a relatively low computational cost, but, if there is high correlation between variables, this method is invalid. The second method depends on the extended range of each attribute. Moreover, it is computationally intensive, and the overall effect of each attribute is prone to be ignored. Therefore, this method is difficult to use for high-dimensional small datasets. The third method does not suffer from the disadvantages of the first two methods. In essence, it obtains the joint distribution between input and output variables from the historical dataset, establishes a mapping relationship between them and the real data distribution, and comprehensively considers the impact of all variables.

## 3. Ultrasound Evaluation Model of the Grain Size of Primary α Phase

### 3.1. Virtual Sample Generation

According to the definition of virtual samples, the generation of virtual samples is divided into two processes. One is to generate $x_{vir}$ through generation relationship K, and the other is to establish a transformation relationship $\hat{H}$ to make $y_{vir}=\hat{H}(x_{vir})$. Because the VSG method based on deep learning can solve the disadvantages of VSG based on sampling and information diffusion, in this study, the deep-learning-based generative model was used to generate virtual samples.

The GAN is a deep-learning model, and it is one of the most promising methods of unsupervised learning on complex distributions in recent years. It is composed of a generator (G) and a discriminator (D). The generator learns the real data distribution, takes the random noise z as the input, tries to fit the real data distribution $P_{data}$, and the output is $G_{z}$. The discriminator identifies whether the input sample is data $G_{z}$ generated from G or real data x. Its structure is shown in Figure 1.

The GAN adopts a min-max objective to train two models with the following objective function:(1)$$\underset{G}{\min}\;\underset{D}{\max}V(D,G)=E_{x∼P_{data}(x)}[\log D(X)]+E_{z∼P_{z}(z)}[\log(1-D(G(Z)))]$$

The learning optimization method of GANs is to fix *G* and optimize *D* to maximize the discrimination accuracy of *D*. Then, *D* is fixed to optimize *G* to minimize the discrimination accuracy of *D*. If and only if $P_{data}=P_{g}$, the global optimal solution is reached, i.e., the generated data can be confused with the real data.

Assuming the real training sample set $D_{s}=(X,Y)$, the steps of generating the virtual sample set $D_{vir}=(X_{vir},Y_{vir})=G(D_{s},N_{vir})$ using GAN are as follows.

(1) The real training sample set $D_{s}$ is preprocessed, and binary coding is performed on it. The accuracy of binary coding is expressed as
(2)$$∆x=\frac{U_{\max}-U_{\min}}{2^{k}-1}$$
where ∆x denotes coding accuracy, $U_{\min}$ and $U_{\max}$ refer to the range of decision variables, and k is the coding length.

Then, the binary coded real training sample set $D_{s}$ is transformed into a black-and-white image set, in which code 1 is displayed as “black” and code 0 is displayed as “white.”

(2) The black-and-white image set is used as the input of the GAN to train the GAN model. The GAN model is saved when $P_{data}=P_{g}$, and the saved GAN model is used to generate a virtual image set.

(3) The virtual image set is decoded and restored to the virtual sample set. The decoding formula is
(3)$$x_{j}=U_{\min}+(\sum_{i=1}^{k}b_{i}\times 2^{i-1})\times ∆x$$
where $b_{i}$ denotes the code of the i-th bit, and $x_{j}$ is the decoded value.

### 3.2. Virtual Sample Validity Analysis and Screening Process

The current virtual sample generation technology can produce a large number of virtual samples, but the validity of the virtual samples is low, and their direct use cannot meet the requirements. Hence, it is necessary to analyze and screen their validity. Taking “the added virtual samples can improve the accuracy of the evaluation model” as the validity criterion, the screening basis is established to screen the generated virtual samples.

The optimization rate (OTR value) indicator is used to measure the validity of virtual samples. The higher its value, the higher the validity of the virtual samples. Equation (4) is the calculation formula.
(4)$$\begin{array}{l} \mathrm{OTR}=\frac{\mathrm{MAP}E_{1}-\mathrm{MAP}E_{2}}{\mathrm{MAP}E_{1}}\times 100\%\\ \mathrm{MAPE}=\frac{\sum_{i=1}^{m}|\frac{y_{i}-{\hat{y}}_{i}}{y_{i}}|\times 100}{m} \end{array}$$
where m is the number of real test samples, $y_{i}$ is the value of the real test samples, ${\hat{y}}_{i}$ is the predicted value of the real test samples, and MAPE is the mean absolute percentage error, which is used to measure the error of the prediction model. The smaller its value, the more accurate the prediction model. Here, $\mathrm{MAP}E_{1}$ is the MAPE between the real test sample value and its corresponding predicted value without adding virtual samples.

Considering that the ultrasound evaluation model is to predict and evaluate the grain sizes of real samples, and the values of the real samples are limited by processing parameters, virtual samples beyond the range of real samples $\left[X_{\min},X_{\max}\right]$ were screened out to avoid introducing unreasonable data.

When verifying the validity of virtual samples, each iteration generates a virtual sample set with a scale of N, in which the virtual sample set generated in the i-th iteration is $D_{vir}^{\prime }(i)=(X_{vir}^{\prime }(i),Y_{vir}^{\prime }(i))=G(D_{s},N)$, $X_{vir}^{\prime }(i)\in [X_{\min},X_{\max}]$. Here, $D_{f}(i-1)$ represents the ensemble of the real training sample set and the valid virtual sample set $D_{vir}^{\prime }(i-1)$, …, $D_{vir}^{\prime }(1)$ generated in the previous iteration. When judging the validity of $D_{vir}^{\prime }(i)$, if $OTR(D_{f}(i))>OTR(D_{f}(i-1))$, then $D_{vir}^{\prime }(i)$ is a valid virtual sample set. Otherwise, it is an invalid virtual sample set. In addition, $D_{vir}^{\prime \prime }=D_{vir}^{\prime }(i-1)\cup D_{vir}^{\prime }(i-2)\cup \ldots \cup D_{vir}^{\prime }(1)$ is the optimal virtual sample set. The pseudo code of its effective virtual sample generation Algorithm 1 is as follows.
**Algorithm 1:** Effective virtual sample generation algorithm (EVSG algorithm)**Input: real training sample set**$D_{s}$**Output: optimal virtual sample set** $D_{vir}^{\prime \prime }$1.**begin**2./* initialize the number of iterations i, initialize reconstructed sample set $D_{f}(0)$, initialize $N_{vir}$, initialize the virtual sample set after initial screening $D_{vir}^{\prime }$*/3. i←04.${\;D}_{f}(0)\leftarrow D_{s}$5.$\;N_{vir}\leftarrow 1$6.${\;D}_{vir}^{\prime }\leftarrow \emptyset$7.  **repeat**
8.  i←i+19.   **repeat**
10.${\;D}_{vir}=(X_{vir},Y_{vir})=G(D_{s},N_{vir})$11.    **if**
$X_{vir}\in [X_{\min},X_{\max}]$ **then**12.${\;D}_{vir}^{\prime }=D_{vir}^{\prime }\cup D_{vir}$13.    **end if**
14.   **until**
$|D_{vir}^{\prime }|=N$
15.${\;D}_{f}(i)=D_{f}(i-1)\cup D_{vir}^{\prime }(i)$16. **until** $\mathrm{OTR}(D_{f}(i))\leq \mathrm{OTR}(D_{f}(i-1))$17. $D_{vir}^{\prime \prime }=\left\{ \begin{array}{ll} D_{vir}^{\prime }(i-1)\cup D_{vir}^{\prime }(i-2)\cup \ldots \cup D_{vir}^{\prime }(1)\; & i\geq 1\\ \emptyset & i=1 \end{array}\right.$
18.**end**

### 3.3. Constructing the Ultrasound Evaluation Model of the Grain Size of Primary α Phase

#### 3.3.1. Ultrasonic Testing Experiment and Metallographic Observation Experiment

TC25 titanium alloy has good high-temperature strength and thermal stability, which makes it an ideal material for aero-engines. Figure 2 is the process diagram of the entire ultrasonic testing test and metallographic observation experiment. The samples in this experiment are titanium alloy ring forgings produced by different process standards. The ring forgings were cut into 168 samples. Metallographic observation samples were prepared, and MR5000 inverted metallographic microscope was used for metallographic observation. More than 20 metallographic images were randomly selected on each sample. Typical metallographic images were shown in Figure 3. All metallographic images of each sample were measured by a digital image processing software, Image J, to obtain the equivalent diameter, area ratio, and grain length/minor axis ratio of all intact primary α phases within the field of view.

For ultrasonic testing, the test surface of each sample is evenly divided into 6 sampling areas of 5 mm × 5 mm, and the ultrasonic testing of the sample is carried out by the contact longitudinal wave echo method. The instruments and equipment used are an Olympus 5077PR pulse generator, a 10 MHz Olympus single crystal straight probe V112-RM, and a Pico Scope 3000 series acquisition card. The size of the probe wafer is larger than the sampling area, which ensures the full coverage of the sample. The nonlinear component of the ultrasonic signals produced by the specimens was acquired using the P-wave collinear harmonic method and a RAM-5000-SNAP (RITEC Inc., Milwaukee, WI, USA) non-linear measurement system. The central frequencies of the transmitting and receiving transducers were 2.5 MHz and 5 MHz, respectively.

TC25 experimental material contains 168 effective samples, and each sample contains five ultrasonic characteristic values (mean sound velocity, mean attenuation, primary offset, secondary offset, and nonlinear coefficient) and one primary α phase grain size value. For the 168 effective samples, the K-fold cross-validation method is used to test the model accuracy, in which K is 3, i.e., 118 effective samples are randomly selected as the training set and the remaining 50 effective samples are used as the test set.

#### 3.3.2. Constructing the Ultrasound Evaluation Model of the Grain Size of Primary α Phase

Each valid sample contains five ultrasonic characteristic values (mean sound velocity ${\bar{C}}_{L}$, mean attenuation $\bar{\alpha }$, primary offset ${\hat{A}}_{F1}$, secondary offset ${\hat{A}}_{F2}$, and nonlinear coefficient $\bar{\beta }$) and the grain size value $\bar{D}$ of one primary α phase. Table 1 shows the partial original data of TC25 materials.

The set composed of five ultrasonic eigenvalues is taken as the input of real samples and expressed in the form of matrix $X_{T}=(X_{1},X_{2},\ldots ,X_{n})$, where $X_{i}=(x_{i1},x_{i2},\ldots ,x_{ir})$, 1≤i≤n. Here, n represents the number of samples $X_{T}$, r denotes the dimension of a single sample, and r=5 here.

The grain size value of one primary α phase is taken as the output of the real samples and expressed as the matrix $Y_{T}=(y_{1},y_{2},\ldots ,y_{n})$. The two together construct a real sample set $D_{T}=(X_{T},Y_{T})$ of TC25 titanium alloy. Because the magnitude of each ultrasonic eigenvalue in the real sample set is different, they are normalized to a unified interval.

The construction of the model is divided into the virtual sample generation process, the virtual sample validity analysis and screening process, and the modeling process. The detailed description is as follows.

(1)Virtual sample generation process.

The TC25 real training sample set is preprocessed and normalized to the [0, 100] interval to ensure the coding accuracy ∆x=0.001. The coding length k=17 is then obtained. A black-and-white image with a scale of 6×17 can be obtained by binary encoding of a valid sample. The image is shown in x (Figure 4), which is the real image data after binary encoding. After preprocessing, $N_{s}$ black-and-white image sets $D_{s}^{\prime }$ can be obtained. Here, $N_{s}$ is the sample set size of the real training sample set $D_{s}$.

The virtual sample image set $D_{vir}=(X_{vir},Y_{vir})=G(D_{s}^{\prime },N_{vir})$ is generated by the GAN, where $N_{vir}$ is the number of a group of virtual sample image sets. The generated virtual sample image set is decoded and restored to a virtual sample set. Figure 4 demonstrates the specific GAN framework, where $G_{z}$ is an output image of generator G.

(2)Virtual sample validity analysis and screening process.

The optimal virtual sample set is generated by the EVSG algorithm, i.e., $D_{vir}^{\prime \prime }=\mathrm{EVSG}(D_{s})$.

(3)Modeling process.

The real training sample set $D_{s}$ and the optimal virtual sample set $D_{vir}^{\prime \prime }$ are combined to form a new reconstructed sample set $D_{f}^{\prime }$, where $D_{f}^{\prime }=D_{s}\cup D_{vir}^{\prime \prime }$. Using $D_{f}^{\prime }$, the support vector regression method was adopted to build the ultrasound evaluation model of the primary α phase grain size.

To verify the prediction model, the prediction accuracy of the real test samples in the model is set, i.e., the test accuracy value $R^{2}$ is the indicator to test the quality of the model. The closer the value is to 1, the better the model. The formula is
(5)$$R^{2}=1-\frac{\sum_{i}^{m}{\left({\hat{y}}_{i}-\bar{y}\right)}^{2}}{\sum_{i}^{m}{\left(y_{i}-\bar{y}\right)}^{2}}$$
where m is the number of real test samples, $y_{i}$ is the value of the real test samples, $\bar{y}$ is the average value of the real test samples, and ${\hat{y}}_{i}$ is the predicted value of the real test samples.

The OTR value and $R^{2}$ value of the model are calculated, and the three processes are repeated M times. The average values of the OTR and $R^{2}$ are calculated.

## 4. Experiment

This experiment was divided into two parts. In the first part, a standard function was used to verify the effectiveness of the proposed method. In the second part, the ultrasound evaluation model of the TC25 primary α phase grain size generated based on the GAN was constructed. The selected datasets were studied in detail, and the experimental results were analyzed.

### 4.1. Analysis of Virtual Sample Effectiveness

#### 4.1.1. Data Description

To verify the effectiveness of the method, a set of benchmark functions was used in the simulation. In this study, a nonlinear benchmark function is considered as
(6)$$y=2.0775+9.04546\times (10^{-1})\times x_{1}+x_{2}^{2}+\cos(x_{3})+1.3556\times (1.5\times (1-x_{4}))+x_{5}^{3}$$
where $x={\left[x_{1},x_{2},x_{3},x_{4},x_{5}\right]}^{T}$ conforms to the uniform distribution of (0,1).

Fifty data points were selected as the original dataset, which was divided into a training dataset (30 samples) and a test dataset (20 samples). Table 2 shows some selected data points.

#### 4.1.2. Parameter Selection

The binary coded training dataset is used to train a GAN to produce reasonable virtual samples. The architecture of the GAN is described in detail below. The generator is a five-layer neural network. Its input is a one-dimensional vector with a length of 100. The three hidden layers are multilayer perceptrons (MLPs) with 512 neurons in each MLP. The generator uses the leaky rectified linear unit (ReLU) as the activation function for all layers except for the last layer, which uses the tanh function for activation. The discriminator uses a four-layer neural network, and the two hidden layers are MLPs with 60 neurons in each MLP. The discriminator uses leaky ReLU as the activation function except that the last layer is activated by the sigmoid function. The generator and discriminator both use the Adam optimizer, and the learning rate of the optimizer is 0.0002.

#### 4.1.3. Analysis of Effectiveness

A standard dataset was introduced to verify the effectiveness of this method. First, the SVM regression model was constructed by using 30 original sample data. Then, the evaluation model based on the GAN was built by employing this method. Finally, the evaluation models established by virtual sample generation methods based on MTD and MD-MTD were compared.

Figure 5 shows the distribution curve of the real and predicted test sample values in the evaluation model with and without virtual samples. As Figure 5 shows, in the evaluation model established by the virtual sample generation method based on MTD, MD-MTD, and GAN, the predicted values of the test sample are closer to the real values than those without virtual samples. The predicted values of the test sample in the evaluation model based on the GAN are the closest to the real values. Therefore, adding effective virtual samples can reduce the error of the model and increase the prediction accuracy of the model. Different virtual sample generation methods have different effects on the evaluation results.

Table 3 shows the MAPE values of the model with and without the addition of virtual samples. After adding virtual samples, the MAPE value of the model decreases. The MAPE value based on the proposed method is the smallest. Hence, this method can verify that adding virtual samples is effective.

Effective virtual samples can fill the information interval between the original sample points and expand the limited and insufficient sample data to enhance the generalization ability of learning and reduce the error of the model.

### 4.2. Impact of Virtual Samples on Ultrasound Evaluation Model

#### 4.2.1. Impact of Virtual Sample Number on Ultrasound Evaluation Model

The virtual sample set is generated by multiple iterations of the GAN, and the impact of the virtual sample number on the evaluation model is observed. Figure 6 shows the curve graph of $R^{2}$ and OTR value of the real training samples in the reconstructed sample set $D_{f}(i)(i=1,2,\ldots )$ (N is 5) as the number of iterations increases.

As Figure 6 shows, with the increase in virtual sample number $N_{vir}$, the curves of both $R^{2}$ and OTR fluctuate with approximately the same trend. When $N_{vir}=35$, both $R^{2}$ and OTR reach their maximum value: $R^{2}$ is 0.834, and OTR is 36.838%. Therefore, the optimal virtual sample number $N_{vir}=35$, and the corresponding model has the best performance.

The OTR value curve indicates that an optimal virtual sample number exists, i.e., there is an optimal subset of virtual samples, which makes the evaluation effect of the ultrasound evaluation model the best.

Figure 7 is a visual diagram of the distribution of virtual sample data and real data generated by the GAN when $N_{vir}=35$. The red point is the virtual sample point, and the blue point is the real sample point. Figure 7a shows the radial visualization, and Figure 7b shows the star coordinate visualization. The red virtual sample points are evenly distributed within the blue sample point area.

#### 4.2.2. Impact of Different Virtual Sample Generation Methods on Ultrasound Evaluation Model

When the impact of different virtual sample generation methods on the evaluation model is analyzed, the number of virtual samples is assumed to be $N_{vir}=35$. Then, the virtual sample generation methods, MTD and MD-MTD ultrasound evaluation (MD-MTDUE), are used to construct the evaluation model and compared with the proposed method. MTDUE is the ultrasound evaluation model based on MTD virtual sample generation method, and MD-MTDUE is the ultrasound evaluation model based on the MD-MTD virtual sample generation method.

For the different virtual sample generation methods, the distribution of test sample values and predicted test sample values in the ultrasound evaluation model is shown in Figure 8.

Table 4 shows the MAPE values of each model under different virtual sample generation methods. Different virtual sample generation methods have different effects on the ultrasound evaluation model. The MAPE value of MTDUE > the MAPE value of MD-MTDUE > the MAPE value of the proposed method.

#### 4.2.3. Comparison with Traditional Ultrasound Evaluation Model

The proposed method was compared with traditional ultrasound evaluation models, such as the single ultrasonic parameter evaluation model, through a series of curve fitting and multi-ultrasonic parameter evaluation models obtained by different machine-learning methods.

The least-squares method was used to establish the ultrasound evaluation model of a single ultrasonic characteristic parameter (sound velocity) and grain size. Here, $\Gamma_{1}$ represents the first-order sound velocity model, and $\Gamma_{1}^{*}$ denotes the second-order sound velocity model. The fitting curve of the single ultrasonic characteristic evaluation model is shown in Figure 9.

Figure 9 shows that a large gap exists between the fitting value of the first-order linear model and the real training value. The second-order linear model fits better than the first-order one.

The ultrasound evaluation model of multi-ultrasonic features and grain size was constructed by a multiple linear regression method and compared with the proposed method. Figure 10 is a contrast curve between the test sample values and predicted values of the test samples in the multi-ultrasonic feature evaluation model. As Figure 10 shows, the curves of the multiple linear regression ultrasonic evaluation model and SVM ultrasonic evaluation model have large fluctuations, whereas the predicted value of the proposed method is closest to the real value of the test sample, and the curve has a small fluctuation range.

Table 5 shows the MAPE values of different ultrasound evaluation models. The single ultrasound evaluation model has a large MAPE value, and its evaluation effect is far lower than that of the proposed method. Hence, compared with the single ultrasonic evaluation model, the proposed method has a better effect. The MAPE value of the multi-ultrasound evaluation model is smaller than that of single ultrasound evaluation model but is significantly higher than that of the proposed method. Therefore, the proposed method is superior to the traditional multi-ultrasonic parameter model.

Compared with the four traditional ultrasound evaluation models, the proposed model has a higher training accuracy, smaller MAPE, and greater stability.

## 5. Conclusions

Small samples have low modeling accuracy because of the limited scale in the ultrasonic evaluation experiment of tissue structure. To address this problem, a method of ultrasound evaluation of the primary α phase grain size based on virtual samples generated by a GAN network was proposed. TC25 titanium alloy forgings were the research object. The experimental conclusions are as follows.

(1) A GAN network was used to generate virtual samples, a virtual sample screening mechanism was introduced, and an ultrasound evaluation model was constructed using SVM regression. The experimental results show that the model has high accuracy and a small error.

(2) The inclusion of virtual samples can better address small-sample problems, such as insufficient sample information or a small number of samples.

(3) Compared with the traditional ultrasound evaluation model, the ultrasound evaluation method with the addition of virtual samples can improve the learning accuracy of the original small samples. Compared with evaluation models constructed using the MTD and MD-MTD virtual sample generation methods, the prediction data obtained by the proposed method are closer to the real sample values. Hence, the ultrasound evaluation model of the primary α phase grain size based on the virtual sample generation by a GAN network has a higher accuracy, a more-stable performance, and less error than other models.

## Figures and Tables

**Figure 1 sensors-22-03274-f001:**
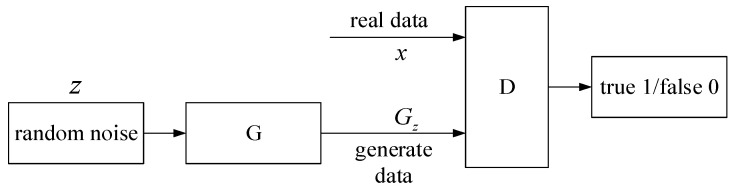
GAN structure diagram.

**Figure 2 sensors-22-03274-f002:**
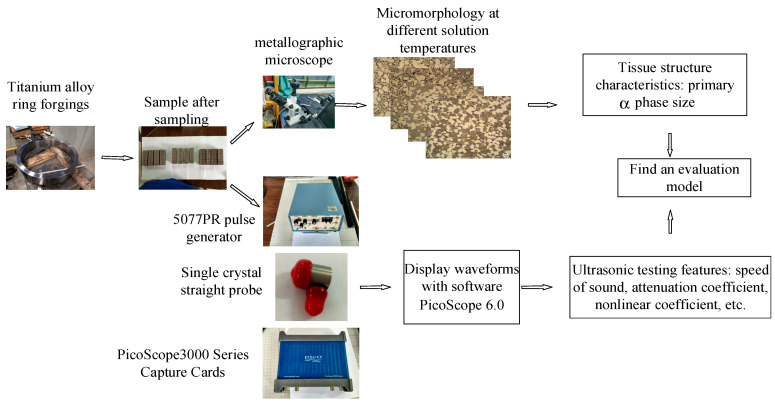
Process diagram of the whole experiment.

**Figure 3 sensors-22-03274-f003:**
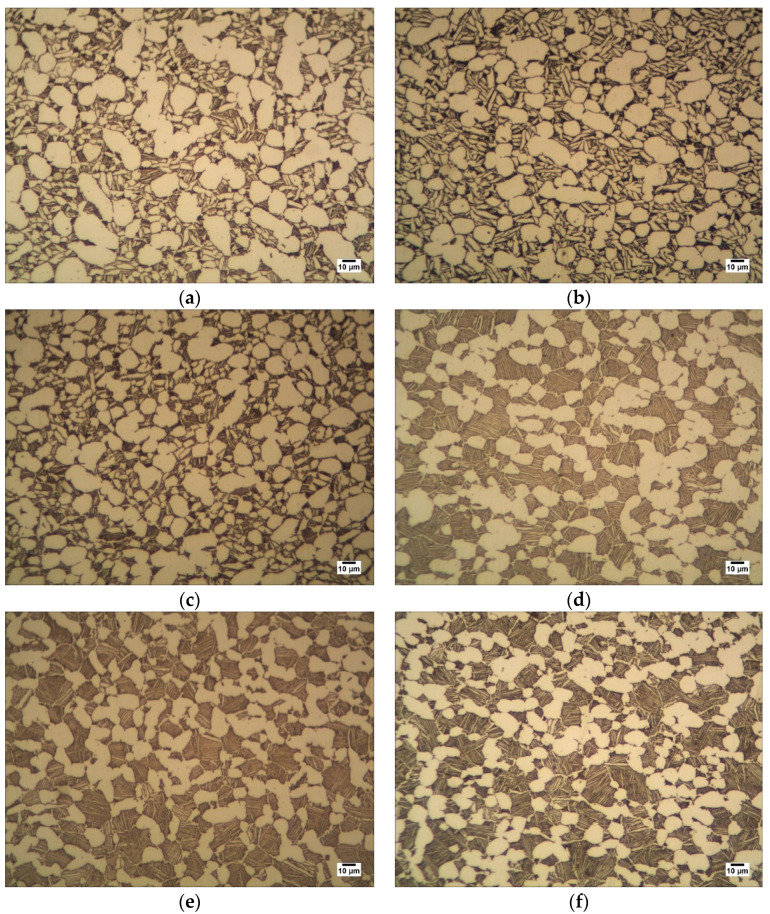
Typical metallographic diagram of ring structure: (**a**) G-1, (**b**) G-2, (**c**) G-3, (**d**) X-1, (**e**) X-2, (**f**) X-3.

**Figure 4 sensors-22-03274-f004:**
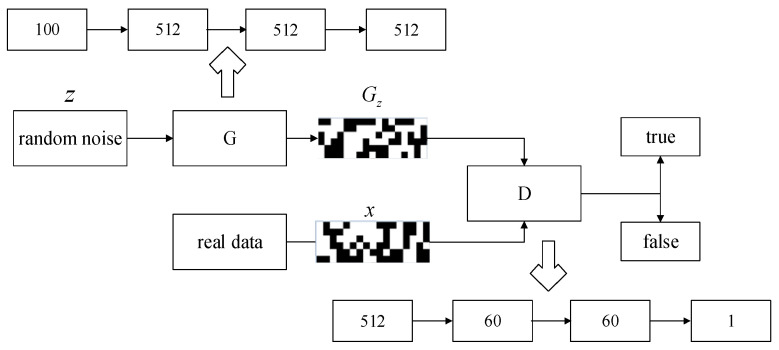
Specific framework diagram of GAN.

**Figure 5 sensors-22-03274-f005:**
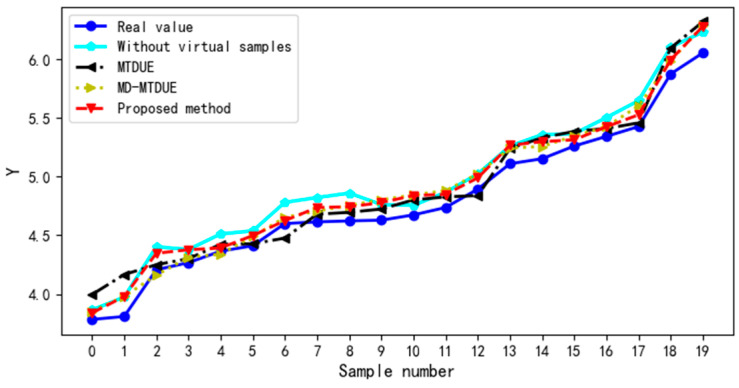
Distribution curve of the real and predicted test sample values in the model.

**Figure 6 sensors-22-03274-f006:**
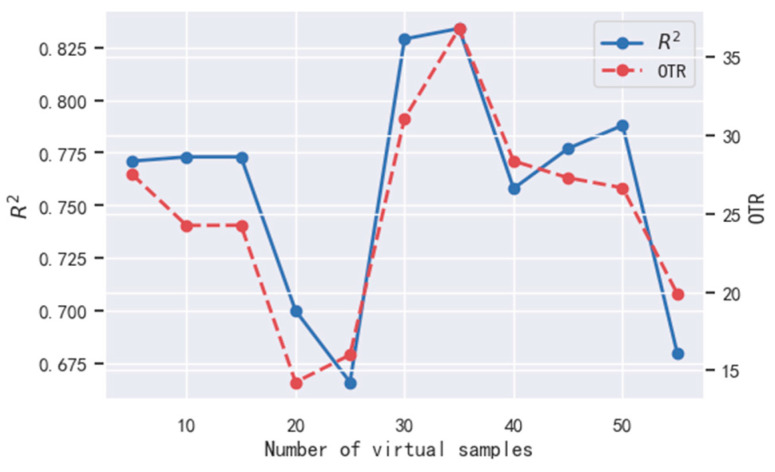
Under different numbers of virtual samples, the *R*^2^ value and *OTR* values of the ultrasound evaluation model.

**Figure 7 sensors-22-03274-f007:**
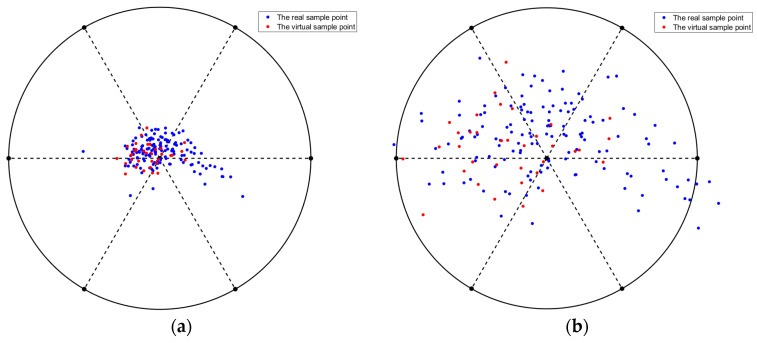
Visualization of virtual sample data distribution generated by GAN network: (**a**) radial visualization, (**b**) star coordinate visualization.

**Figure 8 sensors-22-03274-f008:**
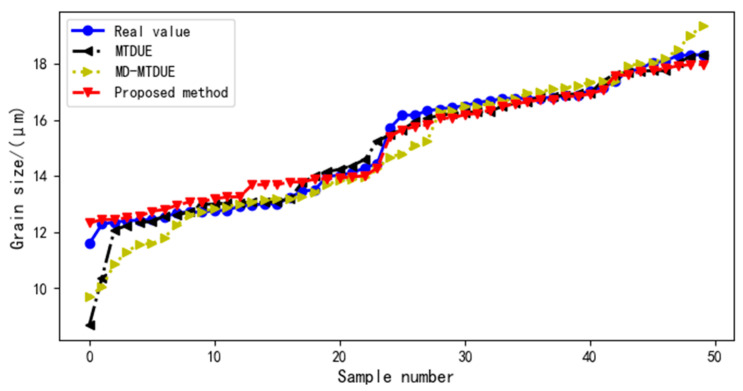
Distribution curve of the test sample value and the predicted value of the test sample in the model.

**Figure 9 sensors-22-03274-f009:**
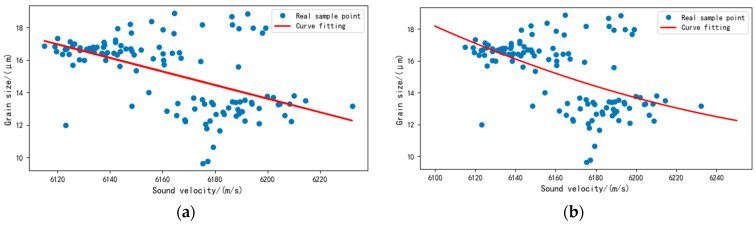
Two single ultrasound evaluation models: (**a**) first-order sound velocity model, (**b**) second-order sound velocity model.

**Figure 10 sensors-22-03274-f010:**
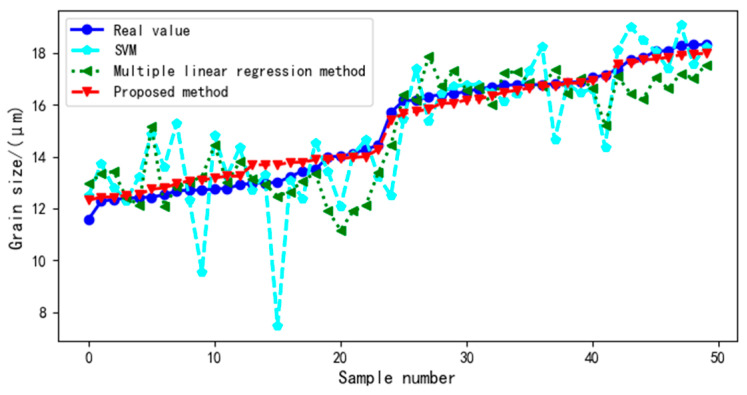
Contrast curve graph between the test sample value and the predicted value of the test sample in the multi-ultrasonic feature evaluation model.

**Table 1 sensors-22-03274-t001:** Partial original data of TC25 materials.

Number	${\bar{C}}_{L}$	$\bar{\alpha }$	${\hat{A}}_{F1}$	${\hat{A}}_{F2}$	$\bar{\beta }$	$\bar{D}$
1	6148.172	0.197	1.351	1.226	3.5	17.670
2	6160.178	0.199	1.268	1.060	3.48	17.858
3	6192.273	0.248	1.309	1.392	4.28	18.831
⋮	⋮	⋮	⋮	⋮	⋮	⋮
n−1	6137.756	0.206	1.5072	1.2791	3.75	17.090
n	6186.24	0.211	1.517	1.351	3.32	18.687

**Table 2 sensors-22-03274-t002:** Partial data of the original dataset.

	$x_{1}$	$x_{2}$	$x_{3}$	$x_{4}$	$x_{5}$	y
Training data (30)	0.957	0.857	0.872	0.837	0.318	4.684
	0.765	0.243	0.805	0.392	0.139	4.761
	⋮	⋮	⋮	⋮	⋮	⋮
	0.033	0.058	0.265	0.210	0.506	4.812
Test data (20)	0.020	0.824	0.573	0.413	0.701	5.153
	0.575	0.310	0.743	0.060	0.099	5.343
	⋮	⋮	⋮	⋮	⋮	⋮
	0.230	0.798	0.623	0.776	0.560	4.366

**Table 3 sensors-22-03274-t003:** MAPE value of the model with and without the addition of virtual samples.

Evaluation Method	MAPE (%)
SVM	3.317
MTDUE	3.05
MD-MTDUE	2.939
Proposed method	2.531

**Table 4 sensors-22-03274-t004:** MAPE values of the model under different virtual sample generation methods.

Evaluation Methods	MAPE (%)
SVM	7.091
MTDUE	6.518
MD-MTDUE	6.357
Proposed method	4.479

**Table 5 sensors-22-03274-t005:** MAPE values of different ultrasound evaluation models.

Evaluation Methods	MAPE (%)
First-order sound velocity method	9.814
Second-order sound velocity method	9.718
Multiple linear regression method	5.607
SVM	7.091
Proposed method	4.479

## Data Availability

Data has been uploaded.

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
