# Peer review of "Ultrasound Evaluation of the Primary α Phase Grain Size Based on Generative Adversarial Network"

_sensors, 2022, doi:10.3390/s22093274_

Round 1

Reviewer 1 Report

TC25 is a TiAlSnZrMo alloy with α + β phases with very good stability at temperatures of 500 degrees Celsius, being used for compressor equipment.

The identification and analysis of the phases of this alloy, using virtual samples and using The ultrasound evaluation model, are the objectives achieved by the authors in this paper.

A generative adversarial network (GAN) is generally used to generate images in areas such as art and video games, the approach proposed by the authors is trying to obtain, using a set of test data, a new set on the same principles.

A single observation related to conclusions that do not highlight the limitations of the applied method and the implications of using this method in the study of the machinability of this type of alloy (TC25).

Author Response

Added clarifications in the text and in the revision notes.

Reviewer 2 Report

The manuscript of the article Ultrasound Evaluation of the Primary alpha Phase Grain Size Based 2 on Generative Adversarial Network by Peng Siqin, Chen Xi, Wu Guanhua , Li Ming, and Chen Hao is devoted to the development of a method for ultrasonic evaluation of the grain size of the primary α-phase in the TC25 titanium alloy, based on generating virtual samples. Ultrasonic evaluation of the grain size of the primary α-phase is a task related to the needs of practice in obtaining information about the average grain size of the alpha phase in two-phase titanium alloys. The average dimensions of the alpha phase in titanium alloys determine their technological deformability and the cutting parameters.

The authors proposed to use the technology of generation of virtual samples with ultrasonic pulse attenuation characteristics similar to the real characteristics of the experimental ultrasonic estimation of grain sizes of the primary α-phase in a two-phase titanium alloy.

Efficient virtual samples were used to expand the set of real samples, and a support vector machine (SVM) regression method was adopted to create a multivariable ultrasonic grain size prediction model.

From the submitted manuscript, it is not clear whether the proposed method allows one to estimate the grain size of the primary alpha phase in two-phase titanium alloys, which include the TC25 (Ti-Al-Zr-Sn-Mo-W) alloy. Grains of alpha and beta phases have different crystal lattices, elastic moduli and sound propagation speeds.

Probably, the proposed technique could be useful in the determination of single-phase alpha or pseudo-alpha titanium alloys. However, the issue of the influence of the concentration and grain size of the beta phase on the recorded parameters of ultrasonic measurements has not been discussed.

Conclusion (3) based on the results of the research does not concern the accuracy of determining the grain size of the alpha phase in a titanium alloy, since the possibility of determining the grain size of the alpha phase in two-phase (alpha + β) titanium alloys has not been proven at all.

 «Hence, the ultrasound evaluation model of the primary  phase grain size based on the virtual sample generation by a GAN network has higher accuracy, more-stable performance, and less error than other models.»

There are no comparisons of the results obtained using the proposed method with experimental data on the grain sizes of the alpha phase for real samples. It is necessary to have reliable data on the distribution of grains in the alloy samples, for example, obtained by the EBSD method with additional data on the distribution of the chemical components of the EDS in the samples.

Comments related to the design of the article.

The introduction should provide information on the structure, chemical and phase composition of the TC25 titanium alloy. In the cited publications [1-4], the structure and properties of other titanium alloys Ti-6Al-4V, TC4, TC11 are discussed.

  1. Line 200. It is necessary to clarify the data in tables 1. How many (n) original data sets obtained during experimental tests were used in the work. The manuscript contains various data. Section 4.1 (Line 271) lists 30 original datasets, parameters for 19 samples are shown in Figure 3. Section 4.2 (Line 295) specifies a sample size of 168 samples. The number of 4 sets of experimental data shown in Table 1 seems to be insufficient to build statistically correct characteristics for the virtual model
  1. Line 257. It should be clarified how the data for the samples in Table (2) were divided. What justifies the division of the number of experimental data in proportion (3:2)? Why don't the amount of data in training data and test data match? Is it possible to determine in advance the total amount of experimental data required to build virtual samples?
  2. It should be clarified which data are indicated in Fig. 8 (Line 368) as the actual values of grain sizes. Are the average values of grain sizes given for the statistical sampling accepted in materials science?

Author Response

Added clarifications in the text and in the revision notes

Reviewer 3 Report

The manuscript (sensors-1636688) «Ultrasound Evaluation of the Primary alpha Phase Grain Size Based on Generative Adversarial Network» Peng Siqin, Chen Xi, Wu Guanhua , Li Ming, and Chen Hao is devoted to the development of an ultrasonic grain size estimation method based on the generation of virtual samples.

The authors proposed an original method for determining the grain size of the primary α-phase using the TS25 titanium alloy as an example.

Ultrasonic evaluation of the grain size of the primary α-phase is a task related to the needs of practice in obtaining information on the average grain sizes of the alpha phase in two-phase titanium alloys.

The main idea of the manuscript is to supplement the results of measuring the attenuation of ultrasound with data for virtual samples with attenuation characteristics of ultrasonic pulses similar to real ones.

Thus, the average grain size of the primary α-phase in a two-phase titanium alloy is proposed to be determined not on the basis of experimental data on the attenuation of ultrasound in samples, but using a combined data set for full-scale and virtual samples.

Efficient virtual samples were used to expand the set of real samples, and a support vector machine (SVM) regression method was adopted to create a multivariable ultrasonic grain size prediction model.

Remarks

  • It is not clear whether the proposed method allows one to estimate the grain size of the primary alpha phase in two-phase titanium alloys, which include the TC25 (Ti-Al-Zr-Sn-Mo-W) alloy.

In this work, a two-phase titanium alloy TC25 was chosen to test the proposed method. The authors did not take into account that the alpha and beta phases have different crystal lattices, elastic moduli, and sound propagation velocities.

  • There are no comparisons of the results obtained using the proposed method with experimental data on the grain sizes of the alpha phase for real samples. It is necessary to have reliable data on the distribution of grains in the alloy samples, for example, obtained by the EBSD method with additional data on the distribution of the chemical components of the EDS in the samples.

The data obtained using the EDS method make it possible to identify grains in the beta phase or grains in a two-phase state. Unfortunately, the authors of the article did not provide data on the distribution of grains of the primary α-phase with which the predictions should be compared.

  • Probably, the proposed technique could be useful in the determination of single-phase alpha or pseudo-alpha titanium alloys.

The influence of the concentration and grain sizes of the beta phase and two-phase grains on the recorded parameters of ultrasonic measurements should be discussed. The possibility of determining the grain size of the alpha phase in two-phase (alpha + beta) titanium alloys should be proven.

4)            Conclusion (3) based on the results of the research does not concern the accuracy of determining the grain size of the primary alpha phase in a titanium alloy.

The possibility of determining the grain size of the primary alpha phase in two-phase (alpha + beta) titanium alloys should be proven.

Notes on the design of the manuscript.

  • The introduction should provide information on the structure, chemical and phase composition of the TC25 titanium alloy. The structure and properties of other titanium alloys Ti-6Al-4V, TC4, TC11, rather than the TC25 alloy, are discussed in the cited publications [1-4].
  • Line 200. It is necessary to clarify the data in tables 1.

How many (n) original data sets obtained during experimental tests were used in this research? The manuscript contains various data.

  • Section 4.1 (Line 271) lists 30 original datasets, parameters for 19 samples are shown in Figure 3. 4) Section 4.2 (Line 295) specifies a sample size of 168 samples.

5) The number of 4 sets of experimental data shown in Table 1 seems to be insufficient to build statistically correct characteristics for the virtual model.

6) Line 257. It should be clarified how the data for the samples in Table (2) were divided. What justifies the division of the number of experimental data in proportion (3:2)? Why does the amount of data in training data and test data not match? Is it possible to determine in advance the total amount of experimental data required to build virtual samples?

7) It should be clarified which data are indicated in Fig. 8 (Line 368) as the actual values of grain sizes. Are the average values of grain sizes given for a statistical sample accepted in materials science?

Round 2

Reviewer 2 Report

Authors addressed most comments mentioned in the previous reviews.

Reviewer 3 Report

The manuscript (sensors-1636688) «Ultrasound Evaluation of the Primary alpha Phase Grain Size Based on Generative Adversarial Network» Peng Siqin, Chen Xi, Wu Guanhua, Li Ming, and Chen Hao is devoted to the development of an ultrasonic grain size estimation method based on the generation of virtual samples.

The authors proposed an original method for determining the grain size of the primary α-phase using the TS25 titanium alloy as an example.

Ultrasonic evaluation of the grain size of the primary α-phase is a task related to the needs of practice in obtaining information on the average grain sizes of the alpha phase in two-phase titanium alloys.

The main idea of the manuscript is to supplement the results of measuring the attenuation of ultrasound with data for virtual samples with attenuation characteristics of ultrasonic pulses similar to real ones.

The quality of the manuscript has improved as a result of the additions and clarifications made.
